# A Community-Based Approach to Integrating Socio, Cultural and Environmental Contexts in the Development of a Food Database for Indigenous and Rural Populations: The Case of the Batwa and Bakiga in South-Western Uganda

**DOI:** 10.3390/nu13103503

**Published:** 2021-10-03

**Authors:** Giulia Scarpa, Lea Berrang-Ford, Sabastian Twesigomwe, Paul Kakwangire, Remco Peters, Carol Zavaleta-Cortijo, Kaitlin Patterson, Didacus B. Namanya, Shuaib Lwasa, Ester Nowembabazi, Charity Kesande, Helen Harris-Fry, Janet E. Cade

**Affiliations:** 1School of Environment, University of Leeds, Leeds LS2 9JT, UK; L.BerrangFord@leeds.ac.uk; 2School of Food Science and Nutrition, University of Leeds, Leeds LS2 9JT, UK; J.E.Cade@leeds.ac.uk; 3Indigenous Health Adaptation to Climate Change Research Team, Kanungu District, Buhoma, Uganda; twesigomwe.sabastian@yahoo.com (S.T.); paulkakwangire@gmail.com (P.K.); didamanya@yahoo.com (D.B.N.); shuaiblwasa@gmail.com (S.L.); esternowembabazi@gmail.com (E.N.); kesandecharity@gmail.com (C.K.); 4School for Policy Studies, University of Bristol, Bristol BS8 1TH, UK; remco.peters@bristol.ac.uk; 5Facultad de Salud Publica y Administracion, Universidad Peruana Cayetano Heredia, San Martín de Porres 15102, Peru; carol.zavaleta.c@upch.pe; 6Department of Population Medicine, University of Guelph, Guelph, ON N1G 2W1, Canada; kpatte08@uoguelph.ca; 7Ministry of Health, Lourdel Road, Nakasero P.O. Box 7272, Uganda; 8Department of Geography, Makerere University, Kampala, Uganda; 9The Global Center on Adaptation, 3072 Rotterdam, The Netherlands; 10Department of Population Health, London School of Hygiene & Tropical Medicine, Keppel St, London WC1E 7HT, UK; helen.harris-fry@lshtm.ac.uk

**Keywords:** food list, food composition database, Indigenous and rural populations, south-western Uganda, community-based research, food culture, environmental changes

## Abstract

Comprehensive food lists and databases are a critical input for programs aiming to alleviate undernutrition. However, standard methods for developing them may produce databases that are irrelevant for marginalised groups where nutritional needs are highest. Our study provides a method for identifying critical contextual information required to build relevant food lists for Indigenous populations. For our study, we used mixed-methods study design with a community-based approach. Between July and October 2019, we interviewed 74 participants among Batwa and Bakiga communities in south-western Uganda. We conducted focus groups discussions (FGDs), individual dietary surveys and markets and shops assessment. Locally validated information on foods consumed among Indigenous populations can provide results that differ from foods listed in the national food composition tables; in fact, the construction of food lists is influenced by multiple factors such as food culture and meaning of food, environmental changes, dietary transition, and social context. Without using a community-based approach to understanding socio-environmental contexts, we would have missed 33 commonly consumed recipes and foods, and we would not have known the variety of ingredients’ quantity in each recipe, and traditional foraged foods. The food culture, food systems and nutrition of Indigenous and vulnerable communities are unique, and need to be considered when developing food lists.

## 1. Introduction

“Zero hunger”, the second of the Sustainable Development Goals, aims to ensure nutritious, diverse and healthy food for everyone [1]. To meet this goal, the United Nations Work Programme for the Nutrition Decade (2016–2025) was developed to promote healthy dietary practices. This includes focus on the implementation of country-specific commitments for action [2] to improve nutrition. Key prerequisites for nations aiming to implement actions to support nutrition are assessment of dietary patterns and nutrient gaps, and the identification of locally available foods that could fill these gaps. To do this, locally relevant food composition tables are required, yet are often unavailable in low-income regions and for vulnerable, marginalized, and Indigenous populations [3].

Existing tools for assessing nutritional intake in populations need to be responsive to different cultural and environmental contexts, especially in the case of Indigenous and rural populations. For example, climate change has a negative impact on food quantity and access, and decreases dietary diversity and quality of foods [4]; in addition, cultural beliefs and social circumstances can affect nutrition and feeding practices [5]. Food items and portion sizes included in locally relevant food composition tables necessarily vary according to the context and population. Detailed information on food lists, such as types, ingredients and recipes using local foods is required for calculating caloric and nutritional intake [6] and for linking foods to nutrient information available from food composition tables. 

Standard methods to estimate food portions [7] and generic food composition tables, are unlikely to be appropriate for Indigenous populations due to their unique cultural and dietary contexts [8]. Food acquisition and preparation are highly influenced by the cultural and environmental context and local traditions, which will influence the development of valid food composition tables [9]. However, the comprehensiveness of national food lists, and accuracy of nutritional composition of mixed dishes is rarely described in the literature, implying limited interrogation of foods which may be missing [10] or different names given to local foods [11].

Here we present a mixed-methods study using a community-based approach for sourcing food and recipe data which considered socio-cultural-environmental factors to construct a food database for the Indigenous Batwa and neighbouring rural Bakiga communities of Kanungu District, south-western Uganda. Prior to our work, the only available food database was designed for central and eastern Uganda [12], and it does not include local foods consumed by the Batwa and Bakiga individuals. 

Inclusion of adjacent Indigenous and non-Indigenous communities allowed us to explore the extent to which cultural constructs of food systems might influence the development of food lists and nutritional assessment methods at the local level. We, therefore, formulated the following objectives: (1) to identify the range of foods and dishes consumed by households (with nutrient composition details in the manuscript of Scarpa et al. [13]); (2) to investigate production, cooking, processing and store methods of consumed foods; (3) to document the factors affecting the food list construction, including food culture, social, environmental, and climatic changes on food consumption, and use of locally-relevant tools to measure portion sizes.

## 2. Materials and Methods

### 2.1. Study Region and Population

The Batwa, who identified themselves as an Indigenous group, and the rural Bakiga people reside in Kanungu District, south-western Uganda. Before 1991, the Batwa primarily lived in the forest and mainly practiced hunting and gathering to meet their nutrition needs. In 1991, they were displaced when the Ugandan Government established Bwindi Impenetrable Forest National Park in the forest where they were residing. Their transition to agriculture has occurred only in the past 20 years [14]. Most of the Batwa in Kanungu District now live in 10 communities on land held in trust by an NGO, the Batwa Development Programme [15]. The Bakiga population, with a longstanding history of agricultural production, represents the majority of the Kanungu District population and resides in the same area as the Batwa. The Bakiga depend on agriculture and livestock, and employ a proportion of Batwa women, and in lesser numbers, men. Farming is now the primary source of income and nutrition for Batwa [16,17], but they are also involved in non-food livelihoods such as tourist performances, brickmaking, charcoal sales, and handicrafts. For both groups, the major crops comprise cassava, beans, millet, bananas, sweet potatoes, and Irish potatoes [18]. 

Both populations are highly vulnerable to the health impacts of climate change and suffer from social and health inequities [17,19,20,21]. Recent survey data indicate that the risks of contracting infections and being malnourished are higher among Batwa compared to the neighbouring Bakiga [15].

### 2.2. Study Design

We used a two-stage process to develop the food database of south-western Uganda for the Batwa and Bakiga communities (Table 1). The first stage, including the generation of food list and recipes, explores the factors affecting food list construction, such as the meaning of food in the local cultural context, social and environmental changes, and the tools to measure portion sizes. In addition, we described how food was cooked, processed and stored. The second stage involves development of an automated online food database [22] for south-western Uganda, and results are reported in Scarpa et al. [13]. 

To create the food list, we used a mixed-method research design with field-based data collection, including focus group discussions, individual dietary surveys and market and shop visits. We use mixed methods to triangulate the data, collecting both quantitative and qualitative data on food and nutrition together with social and environmental contexts [23,24,25,26]. The collection of data was concurrent, and the results from all components were integrated during the interpretation phase [27]. 

### 2.3. Settlement, Participant, and Market Sampling

We used a matched sample of Batwa and Bakiga communities, selecting 4 Batwa communities and the 4 adjacent Bakiga communities (the name of their settlements is the same or similar to the Batwa ones, and the only difference is the adding of the word ‘cell’ to the settlement name). To maximise geographical and culinary variation, we purposively selected pairs of communities according to four strata: close to markets and shops (Bikuto and Bikuto cell), far from markets and shops (Kitariro and Kitariro cell), mid-way between the other two (Kebiremu and Kebiremu cell), and forest-adjacent (Mpungu and Kikome cell). In each of these 8 communities, we conducted one focus group (8 focus groups; *n* = 58 participants in total) and later interviewed the same participants individually. To select the participants, the local researcher met with the community leaders and acquired a list of residing community members. In our sample, we included mothers with children under 5 years, and other women and men of different ages.

The research team also collected data from the four weekly markets in the district, and the top ten most visited shops out of approximately twenty. Although market assessment was conducted by a team member residing in, and familiar with, Kanungu district, five informal conversations conducted with locals helped to confirm the shops where Batwa and Bakiga buy foods most frequently.

### 2.4. Data Collection

The data were collected between July and October 2019 by a local non-Indigenous male researcher from Kanungu District, and female Mutwa and Mukiga (singular for Batwa and Bakiga, respectively) researchers. This facilitated the communication with Batwa and Bakiga chairpersons and male and female participants. Permission for the research was sought from the village chairpersons. Informed written consent was obtained from each individual prior to data collection, with the possibility of withdrawing the consent at any point of the study. Survey tools were pre-tested to ensure they were appropriate and understandable. The dietary surveys and FGDs were conducted in the local language, Rukiga, audio-recorded and later translated into English by the local researcher.

#### 2.4.1. Market and Shop Visits

In the four markets and ten food shops, the research team recorded all foods available using paper forms during a single visit to each market and shop. Of these foods, the type, price, brand, and weight, if indicated, were collected (tool used in Appendix A). 

#### 2.4.2. Focus Group Discussions

We conducted eight FGDs (tool used in Appendix A) with Batwa and Bakiga community members. Through the FGDs, participants had the possibility to discuss and reach consensus about the preparation, processing and storing methods used for foods and dishes, highlighting the potential variability of recipes among the different households. We then investigated factors influencing the generation of the food list with the collaboration of local nutrition experts, which included the exploration of food culture, social and environmental changes affecting food consumption. Additionally, we documented the availability of food in different seasons throughout the seasonal calendar. The FGD guide included open-ended questions only, and the quality of the qualitative study was appraised using the CASP checklist [28]. Power dynamics between research team and respondents were critically examined, and training to the local researchers was given to minimise bias [28], although the local team had experience in conducting FGDs with the Batwa and Bakiga communities. The data saturation (the point in the study where no new information is provided) was reached after interviewing the eight groups. The average length of the discussions was 55 mins.

#### 2.4.3. Individual Dietary Survey

We conducted single-pass 24-hour dietary recall surveys to identify number and types of foods consumed (not amount), using individual interviews with the same individuals participating in the FGDs (tool used in Appendix A). We recorded the number of meals, snacks, and we documented the different tools used to measure portion sizes; also, we collected types of foods and composite dishes consumed on the previous day, including ingredients and recipes. The data collected at this stage helped to confirm foods and recipes collected through FGDs and markets and shops assessment. Dietary information for children under 5 was collected through interviews with their mothers. The dietary survey included open- and closed-ended questions, and the data were audio-recorded and reported on papers. The average dietary survey length was 18 mins.

### 2.5. Data Analysis

Findings of the FGDs were analysed with NVivo Software, and the data presented through quotes and figures. We generated word clouds to provide graphic representations of most commonly consumed food. Any identifiable data were removed to ensure confidentiality.

Thematic analysis, including latent and manifest content, was used to analyse qualitative responses. Latent content analysis is used to extract themes and meanings that are similar in multiple individuals, and this allows for comparison of data. Manifest content analysis is used to count the frequency of topics repeated by multiple individuals [29]. Topics where participants contradicted or discussed different experiences were also reported.

Quantitative data from the individual dietary surveys was analysed using descriptive statistics in Excel® (Microsoft Corporation, Redmond, Washington, DC, USA). Graphs and tables were generated in Excel® and NVivo 12® (QSR International, Melbourne, Australia).

## 3. Results

### 3.1. Study Participants 

The research involved 74 participants, including 58 adults and 16 children (Table 2). The participating mothers gave information on their children’s diet during the individual dietary surveys. All participants invited agreed to participate in the FGDs and individual dietary surveys, except two individuals that decided to be interviewed in the individual dietary survey only. 

### 3.2. Diets of the Batwa and Bakiga Communities

From the focus group discussions, dietary surveys, and market and shop visits, we collected 116 food items and 32 local recipes. When we confirmed the foods collected during the FGDs through the individual dietary surveys, no new items were matched as data saturation was already reached. 

Although women were more likely to cook food for the family and list the ingredients of local dishes when interviewed, men also gave information on alcoholic beverages and other packaged foods consumed outside the house.

Results from the individual dietary survey showed that cereals were the category of foods most consumed on a daily basis (40% of the number of foods reported by Batwa and Bakiga), followed by legumes (32% of the number of foods reported by Batwa; 23% of the number of foods reported by Bakiga) and fruits and vegetables (25% of the number of foods reported consumed by Batwa and 32% of the number of foods reported by Bakiga) (Figure 1). 

Beans were the foods most reported to the interviewers in the individual dietary survey (Figure 2). No Batwa in our sample reported consuming dairy products or eggs the day of recording. Animal proteins were rarely consumed in both populations with lower absolute amount of meat products eaten by the Batwa interviewees. Vegetable proteins (mostly beans) were mostly consumed by Batwa children (43% of the number of foods reported). There was minimal difference in the consumption of fruits and vegetables among adult participants (with higher absolute amounts of fruit and vegetable items consumed by women), with lower amounts consumed by Batwa children (15% of the number of foods reported) (Figure 1). 

Both Batwa and Bakiga women and men during the FGDs and dietary surveys reported a similar number of food items consumed (Figure 1), although Bakiga participants identified slightly higher variety of foods compared to Batwa during the FGDs and dietary surveys (Figure 2). Foods consumed by Bakiga but rarely by the Batwa included more expensive items such as dairy products (e.g., milk and yogurt), insects (e.g., grasshoppers, which are highly seasonal) and fruits (e.g., oranges). 

Some animal-source foods such as meat (goat, pork, chicken) and fish (tilapia, silver fish) were only consumed on special occasions due to their high cost. For example, religious celebrations were the most common days of meat consumption, or when Batwa, and rarely Bakiga, received food aid from governmental and non-governmental organisations (NGOs). In the case of Bakiga, the Government only gives food to the community when elephants from the forest destroy their crops and gardens. 

Alcohol was consumed by both Batwa and Bakiga communities’ members, and according to the women participants “men are used to drink a lot to fill the stomach especially when they are hungry”, and “they buy alcohol instead of buying food for the family”. However, they added that also some women in the communities drink alcohol. ‘Waragi’, locally prepared from matoke and sugar cane, is the most produced and consumed alcoholic beverage in south-western Uganda. 

### 3.3. Production, Cooking, Processing and Storage Methods

The food products consumed by the Batwa and Bakiga populations were typically sourced through subsistence agriculture in small plots/own gardens. Those foods comprised: cereals and tubers such as millet, maize, yams, potatoes, cassava, sorghum; legumes and nuts such as beans, groundnuts, peas; vegetables and fruits such as dodo (leafy greens), matoke, cabbage, tomatoes, onions, avocado, mango, jackfruit, eggplants, watermelon, and carrots; sweets and beverages such as sugarcane and tea. Conversely, animals such as duck, rabbits, and chickens were likely to be bred and then sold in markets to earn enough money to buy foods for the family. 

The most common and cheapest cooking method was reported to be boiling. Frying was reserved for meat and fish, and for tomato sauces, when onions and cooking oil were available. To store food, usually drying was the preferred processing method. Sometimes salt was used to preserve meat or fish, and for seasoning. However, most food (especially fruits and vegetables) was not storable for a long period due to lack of refrigerators. Indeed, foods were kept on shelves at home with other non-edible products. A participant argued that this was another factor affecting the food security in the community: “it is why we do not have enough food” (Batwa FGD). The other Batwa participants in the FGD agreed with this.

### 3.4. Factors Affecting the Generation of the Food List 

See the table below for details (Table 3).

#### 3.4.1. Meaning of Food among Batwa and Bakiga Communities 

The term ‘food’ for the Batwa and Bakiga communities was conceptualised as including a wide range of consumed products. During the FGDs, women described in depth foods and recipes, and men talked more about beverages. It included cereals, legumes, meat, vegetables and fruits, but also herbal drinks and medicinal plants that were frequently consumed by both Indigenous Batwa and Bakiga. Medicinal plants included infusions prepared with local herbs, especially herbs from the forest. Some nutrient-rich vegetables, fruits, cereals, and fish were also considered to have curative properties. For example: kirungi and kazire were factory-bottled herbal drinks believed to treat malaria; jackfruits were consumed to heal stomach pain; roasted yams for diarrheal diseases; greens (green leafy vegetables) to ‘generate blood’; watermelon to cure headaches; honey, milk, avocado and eggplant to heal ulcers; and mudfish to cure worms and malnutrition. Conceptualisations of food also included what were referred to as ‘forest foods’ that were rarely consumed but were still perceived to be part of the culture and sense of community (both Batwa and Bakiga FGDs). These included wild foods such as wild meat, wild yams, honey, or mushrooms found in the Bwindi Impenetrable Park, the forest adjacent to Kanungu District, that the communities are not legally allowed to access.

The term ‘sauce’ was frequently used to indicate any protein-source food that was eaten in conjunction with cereal foods, thus households might refer to a meal of a cereal carbohydrate (e.g., posho) with ‘sauce’ (e.g., beans). In addition, the word ‘greens’ was used to categorize various types of green leafy vegetables, including eggplant leaves, pumpkin leaves, cabbage, okra, Irish potato leaves, guava leaves, amaranth, bean leaves and others.

#### 3.4.2. Seasonal, Climatic, and Environmental Changes

Participants explained when the main staple crops, and other vegetables and fruits were usually harvested during the year, although with climatic changes respondents believed the timings could vary. The seasonal calendar was confirmed through markets and shops assessment (Table 4).

The wet seasons were identified as the best season in terms of availability of fruits and vegetables, however one Bakiga settlement argued that the “dry season (harvesting season) is better because we have food (only crops) saved from the rainy season” Cassava was reported as ‘a resilient crop’ available during both dry and wet seasons (Bakiga FGD). 

The participants noted changing reliability of seasons: “there are no real seasons nowadays” (…) “if rain is there, everything grows, but too much rain can destroy plants (…)”. “Also, if there is too much sunshine, plants will not grow” (Batwa FGD). Batwa and Bakiga interviewees noted that “understanding when to grow crops is very hard, as the weather changes quickly” (Bakiga FGD). According to both Batwa and Bakiga communities, climatic and environmental changes were important factors affecting food availability and quality. Participants discussed that the quality of the soil was decreasing due to over-cultivation and population growth: “people cannot harvest as before during the dry season, and food is decreasing in the forest too” (Batwa FGD). Foods from the forest were perceived to be very nutritious, especially honey, wild meat and greens. “Life is changed, and it does not rain when it is supposed to, thus we no longer have enough food, and we have to go to Bakiga settlements to obtain it. For this reason, we do not have energy and we are sicker, also we have stomach problems” (Batwa FGD). Some participants perceived that the growth of children was compromised due to overconsumption of sweet potatoes, cassava, and beans, without meat or other foods to balance their diet. 

Some participants mentioned that past extreme weather events had a negative impact on foods and diet, contributed to food insecurity and lack of food availability. In addition, they added that due to extreme climatic events crops are destroyed, and foods are not eaten ‘in the right season’, following the time of growth and harvesting of the seasonal calendar. For example, some Bakiga community members recalled: “[In] 1980, when the county was hit by drought and many died of hunger” and “[In] 1999, when rain and storms destroyed gardens, and the Government had to help” (Bakiga FGD). Other participants recalled more recent events: “[In] 2004 and 2017 when there was famine in Kanungu, crops dried up, malnutrition was high, and many children became sick” (Bakiga FGD). 

### 3.5. Dietary Transitions: Forest Displacement and Market Influence 

The displacement of the Batwa from their ancestral home in Bwindi Impenetrable Forest was noted as one of the major causes of diet transitions. To adapt to social and environmental changes, Batwa reported that they “eat less, sometimes once a day”, and “use a new food, maize flour”, that was not produced and consumed in the forest. The use of herbal medicine has also evolved: “we have changed type of foods, portion size and we get sick more easily” and “we go to the hospital when we get sick rather than using herbs from the forest” (Batwa FGD). In addition, food was considered less nutritious: “food does not satisfy (us) as it is not nutritious anymore, does not have enough nutrients” (Bakiga FGD). One Bakiga participant highlighted the high level of food insecurity among the community households: “we have to measure what to eat. In the past, if we had to cook beans, we cooked lots of beans, but now just little beans to protect us from hunger. One kilo of beans now is shared by 8 people and before by 3” (Bakiga FGD). 

The Batwa and Bakiga communities substituted their traditional foods with packaged food purchased from shops and markets: “Nowadays we grow tea and coffee to earn money to buy food, posho and beans (…). We have substituted (our) food because we do not have enough local foods. In the past we used to eat greens and beans without water soup” (Bakiga FGD). Packaged foods bought by both communities comprised bread, posho (produced with millet or maize flour), sweets, soda, honey, rice, cooking oil, pancake, maize, water, sugar, and cassava. 

### 3.6. The Social Context Influences the Number of Meals Consumed

The number of meals consumed by Batwa and Bakiga individuals are represented in Figure 3. Four interviewees reported that those who work in public spaces, such as schools or international companies, often received breakfast and/or lunch in the workplace. In some cases (*n* = 7, including 6 Batwa and 1 Bakiga participant), the interviewees ate the same types of foods for breakfast and dinner or lunch and dinner. Breakfast was the most frequently skipped meal of the day; around half of Batwa and Bakiga men and women did not eat breakfast regularly. 

Dinner was the meal most consumed by both Batwa and Bakiga participants, although 2/18 Bakiga women did not have dinner the day before due to lack of food. Lunch was consumed by all Batwa and Bakiga children and most other participants. Children typically ate at least three meals per day (8/9 Batwa and 6/7 Bakiga children). Some mothers said that they kept food from the day before to ensure their children had three meals per day. 

Generally, participants explained that number and type of foods in meals vary depending on the day of the week, market availability, work starting time, household food insecurity status, and religious events (including fasting): “I am a teacher at school; thus, I eat two meals (breakfast and lunch) when I work, but over the weekend is different” (Bakiga individual interview, male participant).

### 3.7. Tools to Measure Portion Sizes

During the FGDs, the participants reported the most common tools to measure portion sizes: ‘plastic cups’ (500 mL) were used for liquids, ‘kilos’ for cereals and any unpackaged foods available in the market, ‘teaspoons’ (5 g) for oils and ‘soup spoon’ (20 g) for margarine. However, they also added that the family often share meals from the same pan (Bakiga FGD) or plate (Batwa FGD), especially mothers and children, but this is not typical for ‘special occasions’ when each component of the family eats his own food in separate plates. Leftover food is usually eaten by children at the end of the meal, except some ‘richer’ families who may throw it away (Batwa and Bakiga FGDs). 

The focus group participants affirmed that portion sizes changed over the year. The communities reported eating larger portions of foods in the past compared to the current intake. The quantity of food consumed by gender varied between Batwa and Bakiga communities. In the Batwa settlements, men ate more than women and children, according to the participants: “if a man eats a kilogram of food, a woman eats half, and a child ¼”. “This is because men work hard and need more food than women” (Batwa FGD). In the Bakiga settlements there were different, more egalitarian norms, and women or children were equally likely to eat more than men. In Bakiga settlements, men tended to believe that women eat more: “women eat more (1/4 more) than men, and children eat less, but if a child is over 10, he can eat more than a man” (Bakiga FGD), while women reported that children eat more “child eats more because he eats often (…). If there is little food, it is for the child that cries if he is not fed” (Bakiga FGD). 

## 4. Discussion

By using multiple fieldwork approaches, we demonstrated a method for collecting comprehensive data on food, recipes, cooking and storing methods, and factors influencing the development of the food list to construct a locally relevant and contextually specific food database. Our research focused on the Indigenous Batwa and Bakiga communities in southwestern Uganda where this locally contextualised information is lacking, but also provides a general approach that could be adapted method for other contexts. 

In our food list and food database, we included medicinal plants, herbal drinks and traditional foods from the forest, such as grasshoppers and fermented beverages, which were not available in the existing Ugandan database, [12]. The consumption of medicinal plants is common among Indigenous communities [30,31,32] and is part of their culture [33]. Without conducting this fieldwork, we would have missed 27 local recipes, and 6 commonly consumed foods that we included in the online food database. 

Although research on methods for documenting data on food, recipes and portion sizes is growing [7,34,35,36], methods to specifically identify Indigenous dishes and foods remain scarce. Our study highlights the need to include qualitative and community-based methods to obtain more detailed information on commonly consumed Indigenous foods. The lack of Indigenous food databases does not occur only in Uganda, but worldwide; often they are missed in national food composition tables, limiting the assessment of caloric and nutritional intakes in these communities [8]. 

Furthermore, given the trend towards electronic data capture of dietary intakes, it is increasingly important for nutritionists to obtain a comprehensive food list before the dietary assessment begins. The food list is an important input for programming the questionnaires, and incomplete food lists may mean interviewers have to enter missing food items manually or may lead to incorrect entry of similar but different foods.

### 4.1. Key Elements to Be Considered When Collecting Food Lists 

We identified key insights that had important implications for the collection of food lists and development of the food database in this region. 

First, we found that the exploration of nutritional patterns and traditional cooking methods to collect robust local data is difficult without field-based methods informed by anthropological consideration of local knowledge regarding food cultures and peoples’ eating habits [37]. Our findings highlighted the importance of working in collaboration with the community and local researchers to clarify the terminology used to describe foods and recipes [22,38,39]. For example, lack of clarity of the term ‘greens’ used to indicate different green leafy vegetables (as explained by participants and local nutritionists) would have affected the nutritional and caloric intake as it could refer to any unspecified greens available in the food composition table with a different nutritional composition [12]. In addition, this highlights the need to use specific probe questions during the dietary intake assessment stage, to ensure that the required level of details is captured.

Second, we found that understanding not only food culture, but also the meaning of food has implications for mapping the nutritional content of foods, during the food lists collection and the analysis of the results [37]. For example, without understanding the term ‘sauce’ used to indicate the component of a dish rich in vegetable or animal proteins, foods might have been misclassified. A lack of clarity could have resulted in using the nutritional values for any prepared sauce (e.g., tomato sauce) instead of the more precise values for the specific food consumed (e.g., beans as sauce) [12,40] (Table 5): 

In addition, we illustrated the importance of interviewing both male and female participants to include the maximum number of food items and recipes in the food list. As we have shown, men talked more about beverages and women more about foods. In addition, some gender-related issues about the quantity of food usually eaten by men and women in the household raised: male individuals stated that men usually eat less than women, while female participants argued the opposite.

Third, we demonstrate the importance of calculating the nutritional content of recipes at regular intervals to take account of households’ adaptations. Our results indicate that the ingredients of the recipes and cooking methods varied depending on seasonality which influenced the availability of food, and for economic reasons. For example, we found that the recipe for ‘boiled beans’ traditionally was prepared with a minimal quantity of water. However, in highly vulnerable Bakiga and Batwa households there was substantial variation in the bean:water ratio. If the ingredients are not well described, an individuals’ caloric intake would be overestimated (30 kcal / 100g vs 10 kcal / 100g for preparations with less or more water respectively) [12,40].

Fourth, we were able to collect data on culture, society and environment together with the food and dishes consumed in highly food insecure Indigenous and non-Indigenous populations. Only a few studies in literature explored and compared the type of food produced or bought and consumed in Indigenous and non-Indigenous populations, and evidence is especially lacking for sub-Saharan Africa regions [41,42,43]. The Batwa and Bakiga participants reported that they changed diets as they could not hunt or find the same type of food available in the forest thirty years ago. In some cases, the participants turned to purchasing food from markets, substituting traditional dishes with packaged foods with lower micronutrient content. Similar dietary transitions have been documented in many Indigenous communities globally [9,44]. Therefore, Indigenous food lists and databases should be frequently updated as the foods consumed change, and more industrialised and processed foods are eaten. Marketing and advertising promote purchase of highly processed foods such as factory-bottled herbal drinks, that are energy dense products [45] but often low in fibre, vitamin and mineral content [46]. Such highly processed foods are becoming more popular among rural and Indigenous communities in Uganda, and food lists should reflect this aspect.

Lastly, the Ugandan communities affirmed that the quantity and quality of foods changed over time. Climate change is a particular concern for many Indigenous communities as it affects traditional food systems at regional and global level [47]. Climatic changes can contribute to the extinction of wild animals and plant species, and resulting diets, as shown in the research conducted among Canadian Indigenous communities [30]. These changes need to be captured when interviewing the community as foods and recipes can change depending on the availability of foods in a specific time. 

In addition, while the effects of environmental changes on crop yields are widely studied [48,49,50], research assessing the impact on nutritional composition of foods remains limited [51]. Modelling studies have shown the potential for a decrease in food quality due to climate change, and differences in food content between seasons [9,52,53,54]. Our study participants perceived that seasons are different now, and more extreme climatic events are occurring. Consequently, food lists and databases should include the nutritional content of foods over different seasons and in different geographical locations.

### 4.2. Limitations of Our Approach

Some study limitations should be noted. We explored the foods consumed and the cooking and storing methods among the communities together with the local meaning of food, availability of foods and seasonal changes, and tools to measure portion sizes. Although we reached data saturation, possible variations in food preparation methods, and new processed or commercial food products could be missed [37]; continued collaboration with the community and local nutritionists is needed to identify new foods and composite dishes that are introduced over time. We could not find the nutritional information of seven commonly eaten traditional plants in any of the other West African databases (the food composition table for central and eastern Uganda, the Tanzania food composition tables [55] and the Kenya food composition tables [56]). This limitation can be further addressed by conducting laboratory analyses of Indigenous foods. 

Our work included a seasonal calendar and detailed dietary recalls, but we limited our fieldwork to the harvesting season, therefore more variation may be found in other months of the year. Additionally, finding the most appropriate tools to measure portion sizes among Batwa and Bakiga was difficult because families often ate from the same plate; estimation of leftover foods was also complex, although not very common as food was usually given to the children at the end of a meal. Further work is needed to develop accurate and ‘easy-to-use’ tools to measure food portions, such as the use of video-cameras [57] to quantify eaten foods in remote areas.

## 5. Conclusions

Despite some limitations, the data resulting from the study were essential to develop the food list for south-western Uganda. We used a community-based method for collecting rich information required to develop an Indigenous food list by taking into account the social, environmental and cultural factors influencing the food lists construction. Our study design—including FGDs, individual dietary surveys and shop and market visits—enabled development of a comprehensive assessment of Batwa and Bakiga food consumption and diet. These methods can be used in other Indigenous communities, although some tailoring to the social, environmental and cultural context would be required. Detailed and culturally-relevant food lists will improve the calculation of individual food and nutrient intake, which will enable policy makers, international and governmental organizations to adapt nutritional interventions to the specific community needs [58]. This will aid work aiming to improve Indigenous peoples’ nutritional status, and achieve the sustainable development goal of “zero hunger” [1].

## Figures and Tables

**Figure 1 nutrients-13-03503-f001:**
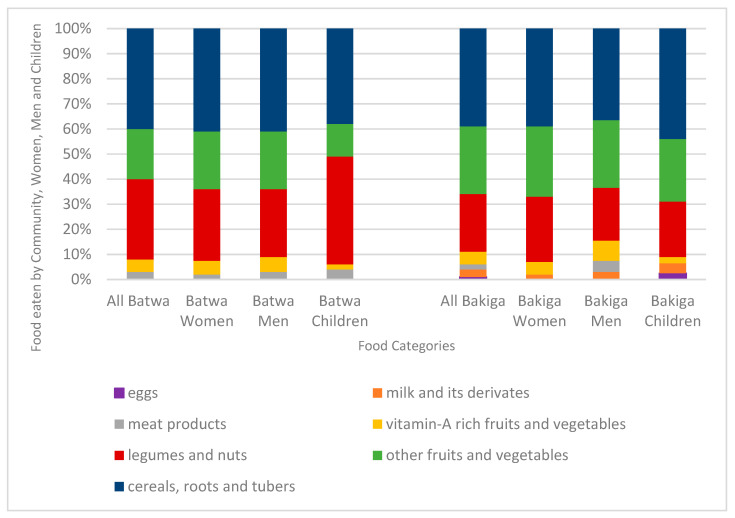
Percentages of frequency of different food groups eaten by the Batwa and Bakiga participants (women, men and children) in the previous 24 h (data collected through the individual dietary survey).

**Figure 2 nutrients-13-03503-f002:**
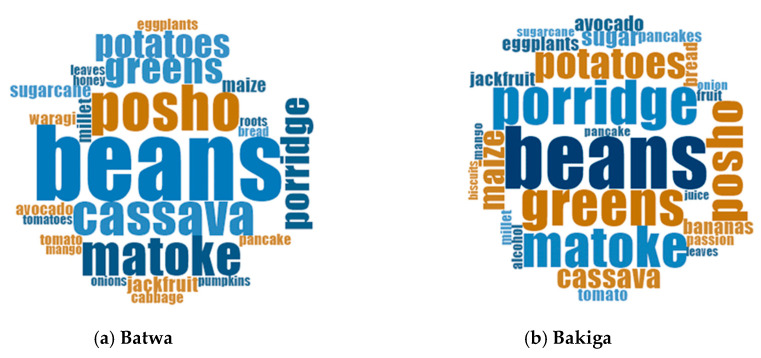
(**a**,**b**). Word frequency analysis to explore the most eaten foods among Batwa (**a**) and Bakiga (**b**) participants. The bigger the word size, the more times the food item was mentioned during the individual dietary surveys (Scale: 1–50 words). Beans were the foods most commonly eaten in both communities together with matoke (type of banana). Posho and cassava were more consumed among the Batwa, while greens and porridge among the Bakiga.

**Figure 3 nutrients-13-03503-f003:**
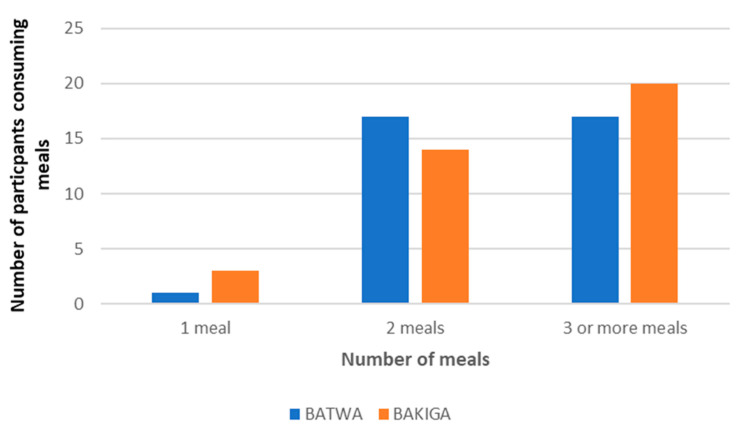
Number of meals among Batwa and Bakiga participants.

**Table 1 nutrients-13-03503-t001:** Methods used for local, field-based sourcing of food and recipe data.

Steps	Methodology	Purpose
1.1 Generate list of foods and collect recipes	Shop and market assessment	To characterise foods and brands (if available) sold in the shops and markets.
	Focus group discussions	To document the foods and recipes consumed by the communities
	Individual dietary survey	To confirm the foods consumed and ensure saturation of data for the food list
1.2 Document cooking, processing and storage methods	Focus group discussion	To characterise cooking and storing methods, and document condiments used for local recipes/forest foods (composite dishes)
	Individual dietary survey	To identify missing cooking and storing methods of foods and recipes and foods given to children
1.3 Document factors influencing the food list construction	Focus group discussions	To list foods available in the dry and wet season (seasonal calendar) and consumed during the weekends and holidays/special occasions To investigate food culture, the meaning of food, and socio-economic and environmental factors
	Individual dietary survey	To explore tools to measure portion sizes

**Table 2 nutrients-13-03503-t002:** Number of participants in the focus group discussions and individual dietary surveys.

	Focus Group Discussions	Individual Dietary Surveys *
Batwa participants	27 Batwa individuals interviewed: Community 1: 6 individuals Community 2: 8 individuals Community 3: 5 individuals Community 4: 8 individuals	Women: 14 Men: 12 Mothers’ reports for children (under 5): 9 Declined to participate: 1
Bakiga participants	31 Bakiga individuals interviewed: Community 1: 7 individuals Community 2: 8 individuals Community 3: 8 individuals Community 4: 8 individuals	Women: 18 Men: 12 Mothers’ reports for children (under 5): 7 Declined to participate: 1

* Individual dietary surveys were conducted with mothers and children from the same household, but men were from different households.

**Table 3 nutrients-13-03503-t003:** Main qualitative findings of our study with associated quotes collected during the Batwa and Bakiga focus group discussions (FGDs) and individual dietary surveys.

Factors Affecting the Food List Construction	More Details from Quotes
Importance of food culture and meaning of food when collecting food lists.	“We call ‘sauce’ the foods that accompany cereals, for example beans” (Batwa FGD, male participant) “Forest foods (including medicinal plants) are part of (our) culture and (our) sense of community” (Batwa FGD, female participant)
Food type and consumption change overtime, and they are influenced by climatic, environmental and demographic changes.	“People cannot harvest as before during the dry season, and food is decreasing in the forest too” (Batwa FGD, male participant) “Life is changed, and it does not rain when it is supposed to, thus we no longer have enough food (...)” (Batwa, male participant) “Children eat only sweet potatoes, cassava and beans, and (children) do not grow like before” (Batwa FGD, female participant) “If rain is there, everything grows, but too much rain can destroy plants (...)”. (Batwa FGD, male participant) “Also, if there is too much sunshine, plants will not grow” (Bakiga FGD, female participant)
Dietary transitions due to displacement and market influence	“We eat less, sometimes once a day” (Batwa FGD, male participant) “We use a new food, maize flour” (Bakiga FGD, male participant) “We have to measure what to eat. In the past, if we had to cook beans, we cooked lots of beans, but now just little beans to protect us from hunger” (Batwa FGD, female participant) “We have substituted (our) food because we do not have enough local foods. In the past we used to eat greens and beans without water soup” (Batwa FGD, female participant)
Production, cooking, processing and storage methods limit the consumption of certain types of foods during the dry and wet season.	“We usually grow foods in our gardens, and we usually sell animals to get enough money to buy food for our family” (Batwa FGD, female participant) “We boil most of our foods, and we use onions and Kimbo (cooking oil) only when we have money” (Batwa FGD, female participant) “Most of our food is not storable: we don’t have refrigerators and (this) is why we do not have enough food” (Batwa FGD, male participant)
Frequency of meals and foods type consumed is linked to social context	“I am a teacher at school; thus, I eat two meals (breakfast and lunch) when I work, but over the weekend is different” (Bakiga Individual Interview, male participant) “We usually eat meat during Christmas, but not always. Only if we have money” (Batwa FGD, female participant)
Tools to measure portion sizes vary among individuals and are influenced by social and cultural context.	“The family usually eats from the same plate” (Batwa FGD, female participant) “One kilo of beans now is shared by 8 people and before by 3” (Batwa FGD, male participant) “We use plastic cups for porridge” (Bakiga FGD, female participant) “Women eat more (1/4 more) than men, and children eat less, but if a child is over 10, he can eat more than a man” (Bakiga FGD, male participant) “Child eats more because he eats often (...). If there is little food, it is for the child that cries if he is not fed” (Bakiga FGD, female participant)

**Table 4 nutrients-13-03503-t004:** Seasonal calendar with the most eaten crops, fruits and vegetables according to the Batwa and Bakiga participants. Lightening (wet season) and sun (dry season) icons indicate when the food items are harvested and eaten.

Type of Food (Common Names)	Wet Season(Mid-January-End of February)	Dry Season (End of February-Mid May)	Wet Season (Mid-May-Mid August)	Dry Season (15th Mid-August-15th Mid-January
Cassava		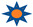		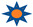
Maize				
Millet				
Beans				
Sweet potatoes		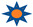		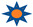
Irish potatoes				
Cabbage				
Mango		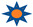		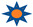
Jackfruit		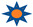		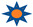
Pineapple		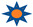		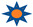
Green leafy vegetables				
Groundnuts				
Yams		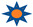		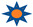
Eggplants		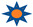		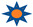
Watermelon		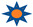		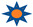

Lightening (wet season) and sun (dry season) icons indicate when the food items are harvested and eaten.

**Table 5 nutrients-13-03503-t005:** Nutritional values of tomato and bean sauces.

100 mL of Each Item:	Kilocalories [12,40]	Proteins [12,40]	Lipids [12,40]	Carbohydrates [12,40]
Tomato sauce	68 kcal	2.0 g	5.0 g	6.0 g
Beans sauce	81 kcal	5.7 g	0.2 g	14.6 g

## Data Availability

Data is contained within the article, and no further information about qualitative data can be shared due to ethical/privacy reason as we worked with vulnerable communities.

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
