# Peer review of "A Community-Based Approach to Integrating Socio, Cultural and Environmental Contexts in the Development of a Food Database for Indigenous and Rural Populations: The Case of the Batwa and Bakiga in South-Western Uganda"

_nutrients, 2021, doi:10.3390/nu13103503_

Round 1

Reviewer 1 Report

I read the manuscript with a great interest mainly due to a lack of knowledge of this culture. The problem included in it is of great importance in assessing the way people in low income regions and the vulnerable, marginalized, and Indigenous populations eat. The fact that certain foods are not included in the food composition tables means that they cannot be included in the evaluation, even though foods are consumed in the large amount.

This approach was used in the Introduction, while the manuscript takes into account a wider scope, np. the factors affecting the food list construction, including food culture, social, environmental, and climatic changes on food consumption. In this situation, the title of the manuscript, introduction and the aim need to be thought over and changed. Moreover, the aim describes rather the scope of the research  than its aim.

I assess positively a few methods to obtain the data, i.e. group discussion, 24-hour interview, market / shop visits. However, the problem is the participation of the same people in the group discussion and interviews. Please explain it.

Although the authors indicated that the aim of using the individual interviews was to confirm and validate the information obtained in the group discussion, any information about the result of this action was nowhere to be found.

In my opinion, a group discussion is not a good method of obtaining information about recipes. Such information would be better obtained in individual interviews - as recipes may differ from one family to another. Therefore, one could ask about the recipes of the dishes indicated in the discussion, and then compare the compositions.

More detailed information about market / shop visits is needed – e.g. a single visit? Also, more detailed information about informal conversations with locals which helped to identify the shops where Batwa and Bakiga buy foods most frequently is needed.

In my opinion, the results of each study (discussion, 2-4-hours recall, shop visits) should be presented separately, and then a comparison of them.

If the aim of the study was to list foods that have not been included in the tables so far, in my opinion, part of the manuscript "The social context influences the number of meals consumed" goes beyond the title of the manuscript. It is important what is eaten not when it is consumed.

In conclusions Authors stated that they developed a simple method for collecting  rich information required to develop an Indigenous food list by taking into account the social, environmental and cultural factors influencing the food lists construction. However, it was not the aim of the study. Moreover, such a method should be described in detail in the manuscript (sample, methods), not in supplements, and then verified.

Reviewer 2 Report

This is an instructive paper, clearly written and scientifically sound. It has some elements of novelty such as the foods consumed change, and more industrialized and processed foods are eaten, so Indigenous food lists and databases should be frequently updated. Detailed information on food lists using local foods is required for calculating caloric and nutritional intake and for linking foods to nutrient information available from food composition tables, which will improve the calculation of individual food and nutrient intake. It is a research study that highlights how important this information is for international and governmental organisations be able to adapt nutritional interventions to the specific community needs.

However, the manuscript requires some minor corrections before it is ready for publication. The list of suggested changes follows.

DETAILED COMMENTS

  1. In the abstract (line 26), the word "discussions" is missing after “focus groups” for a correct designation of (FGDs), as have been explained in line 108 in study the methods of the 2. Study design (Materials and Methods).
  2. In the keywords (line 36), the word “Food” in "Food list" and "Food composition database" should be written in lowercase letter, all other keywords are in lowercase. In the word "south- western" the space between the hyphen and western should be removed.
  3. The headings "2. Material and Methods" (line 77) and "4. Discussion" (line 385) should be in bold, as are the headings: 1. Introduction (line 39), 3. Results (line 185) and 5. Conclusions (line 497).
  4. The heading "2.1. Study region and Population" (line 78) should be in italic font, as are the similar headings.
  5. In section "2.1. Study region and Population" (lines 78-97) the population of Batwa and Bakiga should be characterized by age and sex if it is known.
  6. In Table 1, in the Step “1.1 Generate list of foods and collect recipes” the “Purpose” indicates “To confirm and validate foods consumed”. I don’t see a validity of the food records has been evaluated. If I am right, “validate” should be removed as a purpose from the table, as indicated in lines 166-168: The data collected helped to confirm foods and recipes.
  7. It should be used the singular of the word “brands” in line 142. Other words in the same sentence are also in singular form.
  8. Lines 156-157 need a reference.
  9. In Table 2, in the column "Focus group discussions" must be together  “com” and “munity”, parts of the word community.
  10. In Table 2, the explanation of 1 is missing in “others’ reports for children (under 5)1: 9” and “others’ reports for children (under 5)1: 7”
  11. Move Lines 205-206 before Figure 1.
  12. Lines 210-212. If quantitative data from the individual dietary surveys have been statistic analyzed should be indicated, and if it is the case in results in Figure 1 too.
  13. Line 397. After “and 6 commonly consumed foods”, please add a list of names starting with “such as…”.
  14. Between lines 431 and 432, is it a Table? If this is the case, better to indicate as Table 5.
  15. In this Table the unit Kcal should be written in lowercase: kcal.
  16. After Line 543, Appendix C (in a separate word file as requested) is missing.
  17. In References: 1, 2, 44, 46, 54, the format should be homogenised: [cited year date].
  18. In Reference 28: the space between CMAJ and the colon should be removed (Line 588).
  19. In Reference 28: “journal” in “Canadian Medical Association Journal” should be written in capital letter (Line 589).
  20. Reference 40: “nutricion” in “Archivos latinoamericanos de nutrición” should have accent mark.

Round 2

Reviewer 1 Report

I appreciate all changes made in manuscript.  Now the reading and understanding of the text is much easier.

Kind regards

Reviewer